# An Investigation into the Effects of Changing Dorso-Plantar Hoof Balance on Equine Hind Limb Posture

**DOI:** 10.3390/ani12233275

**Published:** 2022-11-24

**Authors:** Yogi Sharp, Gillian Tabor

**Affiliations:** 1Lion House, Lion Hill, Stone Cross, Pevensey BN245EG, UK; 2Equestrian Performance Research Centre, Hartpury University, Gloucestershire GL193BE, UK

**Keywords:** hind hoof balance, posture, proprioception, therapeutic farriery, physiology

## Abstract

**Simple Summary:**

A link between hind hoof balance and pathologies in the hind limb of horses is emerging. However, the timeline of causation remains elusive, meaning a lack of treatment protocols within the veterinary and farriery industries. One area suggested as a link between the hind hoof balance and pathologies is hind limb posture. This study aimed to test the theory of the hoof being a neuro-sensory organ responsible for informing equine stance and the suggestion of a functional link between the hoof balance and hind limb posture. All horses presenting with negative plantar angles, a severe form of long toe, low heel conformation, also presented with a canted-in posture which was changed by farriery intervention. The implications for practice suggest that both the hind hoof balance and the pathologies in the hind limb and trunk could be resultant from the presenting posture, therefore showing the importance of a multi-profession approach to managing both hoof balance and higher pathologies, and the importance of correction in the treatment of postural dysfunction and pathology within the hind limb and trunk. Postural assessment should become part of farriery protocol and be incorporated into intervention decisions.

**Abstract:**

Links between poor hind hoof balance, pathologies in the hind limb and associated altered posture have been suggested but not quantified. The hoof is proposed as a neuro-sensory organ responsible for informing equine stance with implications for musculoskeletal health in the hind limb and trunk of the horse. This study aims to quantify equine limb posture and its relationship with hoof balance. Twelve horses presenting with negative plantar angles were photographed and limb posture documented before and after the creation of positive plantar angles and improved three-dimensional proportions around the centre of rotation of the distal-interphalangeal joint, using farriery prosthetics. The results showed that horses presenting with negative plantar angles had canted-in postures and that farriery intervention had a significant effect on hind limb orientation in seven of these horses. There was a significant difference in metatarsal angle pre and post intervention with the mean for pre intervention being 81.3° ± 5.1 and post intervention being 88.0° ± 3.8 in the right hind and 74.4° ± 3.7 and 87.1° ± 2.9 in the left hind. The findings of this study support the hypothesis that the hoof balance informs equine stance and can play a role in affecting limb posture.

## 1. Introduction

The associations between hoof conformation, lameness, degenerative disease, and catastrophic injury have been widely documented [1,2,3,4,5,6,7,8,9], with studies into the relationship between long toe and low heel conformation in the fore limb, showing strong associations with pathology within the digit, notably navicular syndrome [3,6,7,8]. In the hind limb, there are reports of the association between hoof balance and concurrent pathologies, seemingly referring to higher structures [2,4,5]. These studies have suggested kinematic research examining altered biomechanics is needed, to elucidate the causation. However, authors have also alluded to a canted-in postural adaptation (Figure 1) in horses presenting with poor hind hoof balance but have not quantified or elaborated further on this point [2,4,5].

Although these previous studies only mentioned posture as an observation, it has been suggested that posture is the important link and possible reason for referred pathology, with altered posture indicated as a possible cause of changes in hind hoof morphology [5]. In support of this relationship, studies have outlined the hoof as a neurosensory organ [12] responsible for proprioceptive feedback that informs equine stance. However, as with postural relationships, this effect has not been quantified, therefore greater knowledge of the link between hoof balance and posture could begin to elucidate understanding of these wider relationships.

### 1.1. Posture

Whilst conformation is the length and shape of bones, posture is how the horse orientates and supports those structures. Limb orientations, such as canted-in, cow hock and base narrow have been measured as conformation markers, with a scoring system [13]. However, recent studies and anecdotes within the industry, have raised questions on the accuracy of this definition, terming it as an Abnormal Compensatory Posture (ACP) [2,4,5,10,11]. Conformation has been linked to musculoskeletal injury in racehorses [14], has been associated with locomotor health and sports performance [15] and hind limb conformation has been linked to proximal suspensory desmitis [16]. However, studies into the effect of equine posture on soundness and performance are limited and have focused on the head, neck and spine [17,18,19]. Two groups have proposed objective measurement techniques [17,18,19] enabling quantification of the head–neck–back relationship with other variables. This system does not include limb orientation, nor is it addressed in the literature by any other quantification than as conformation [13]. Studies have outlined proposed causality and the ability for equine management and physiotherapy treatment to affect posture in the head, neck and trunk [17,18,19], therefore also establishing them as postures rather than conformation. However, small attention has been paid to the limb orientations associated with ACP. Emerging research into the significance of limb orientation has been limited to clinical observation and a mathematical model [10], which suggests that canted-in metacarpal/metatarsal alignment requires increased neuro-muscular effort to maintain stability. While the effects on stability have been quantified in base narrow stances [20], showing increased muscular activity and postural sway, recognition of and research into the mechanical and physiological effects of canted-in limb posture is limited. 

Distorted neural signalling from three main proprioceptive regions is associated with ACP [21]: dental occlusion and the temporomandibular joint (stomatognathic system), the upper cervical muscles (poll) and hoof balance. However, while preliminary experimental studies (Gellman, Shoemaker and Reese, unpublished data) have documented changes from ACP to Normal Neutral Posture (NNP, defined as limbs with vertical metacarpals/metatarsals), after normalizing these three structures and functions [11], no studies have isolated the influence of hoof balance alone. 

### 1.2. Negative Plantar Angles

Recent studies indicating that change in posture is associated with poor hind hoof balance have focused on negative plantar angles [2,5] and their link to pathologies in the hind limb. Negative plantar angle (NPLA) is defined as a negative angle to the distal phalanx (Figure 2).

NPLA is an extreme presentation of long toe/low heel hooves, recently documented as the most common hind hoof pathology [22]. Negative palmar angles (NPA) in the forelimb [23], are created by a progressive collapse of the heels and are graded according to severity (Table 1).

**Figure 2 animals-12-03275-f002:**
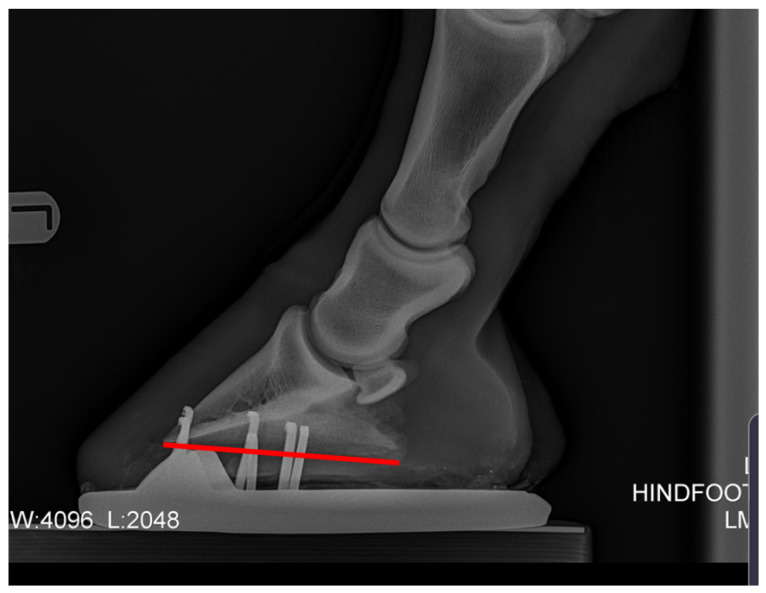
Negative plantar angle takes its name from a principal radiographic feature—the solar or plantar margin of the third phalanx (P3) has a negative angle in relation to the ground surface, and sole depth under the dorsal-distal margin (tip) of P3 is greater than that under the palmar processes (wings) when viewed on a lateral radiograph [23].

The progression from long toe/low heel to NPLA, due to the canted-in posture, has been suggested (5), however, other variables must be considered. Some studies [24,25,26] suggest that hoof growth between trim/shoeing cycles can result in sufficient morphological and mechanical changes to promote development of NPA/NPLA. One author [23] suggested that grade 1 NPA could be addressed with appropriate trimming, postulating that incorrect trimming could exacerbate the morphology and lead to more severe grades of NPA. With more severe cases of NPA, a positive palmar angle remains unachievable without mechanical shoeing [23]. Anecdotally, the same is true for the hind limbs. This supports the suggestion of static and dynamic forces acting on the hoof to create the perpetuating negative morphology, as changes in hoof morphology are linked to changes in magnitude and direction of forces acting on them [27]. 

NPLA and the relationship between hoof balance/proportions and limb orientation are worth investigating for welfare and performance optimisation. The whole horse is connected from head to toe; therefore, changes in forces, load and strains on the hind hoof are not limited to the foot, but affect the entire leg, and presumably the rest of the body.

The aim of this study was to establish a correlation between hind hoof imbalance in the dorso-plantar plane and canted-in hind limb position, using metatarsal angle as a quantitative indicator. Specifically, this study hypothesised that farriery intervention would create a positive plantar angle and external hoof-pastern axis alignment, therefore affecting metatarsal position. 

## 2. Material and Methods

The proposal for the study received ethical review by a university institution prior to the farriery work being undertaken. The horses were referred for intervention by their veterinary surgeons. 

A convenience sample of twelve horses of different breeds, ages, height, and weights (available horse information is presented in Table 2) with previous radiographs confirming NPLA by veterinary diagnosis, presented to the primary author (YS) for second opinion farriery recommendations, were included in the study. Seven of the horses had a treatment method agreed between the author (Qualified farrier, Diploma of the Worshipful Company of Farriers, BSc (Hons)) and veterinary surgeon as part of standard practice protocols in the best interest of the clinical presentation. The NPLA grade was assessed by the author with agreement from the referring veterinarian. Radiographs were taken by a variety of referring veterinarians according to their practices protocols and the protocols were unseen by the author. Farriery intervention and data collection were performed within one week of radiograph acquisition. Two of the horses were outside of the one-week timeframe (horses 11 and 12); therefore, their radiographic measurements were not included in analyses. The horses were positioned for postural photographs by walking four strides and then stopping with both front feet next to each other, after standing for 1 min, they were then photographed from the left and right sides. A camera (iPhone 11, 12 megapixel) was used to take a lateral photograph of the hind limb centred perpendicular to the region of the distal biceps femoris/proximal extensor digitorum longus, from 3 metres (Figure 3). 

A shoeing protocol for treating NPA [23] was adapted and modified for the hinds by the primary author (YS). The trim looked to establish improved plantar angle toward positive, by removing sole and horn from the anterior (toe) region with guidance from the radiographs and palpation of the sole, to determine proximity to the solar dermis. Any toe flare (distorted wall) was removed to the extent possible, without compromising the integrity of the dorsal wall. The dorsal wall was maintained at the same angle from the first 3 centimetres distal of the coronet to ground unless there was prior rasping by previous farrier. As little heel height as possible was removed to maintain plantar angle, but it was ensured that the bearing surface at the heels had any collapsed horn trimmed to create tubules orientated perpendicular to the ground (Figure 4). 

If heel removal beyond the plane of the quarters was necessary to remove collapsed horn, the solar surface was trimmed on two planes and a gap was created between the heels and the shoe as the shoe package rested on the prolapsed frog. The shoe was fitted to provide reduced breakover distance [4] by fitting to a line dropped from the bulbs of the heel and shoeing under the toe and blunting back the toe perpendicular to the ground, to create a distance from the most dorsal distal point of the solar corium to the junction of the dorsal wall and sole radiographically equal to the thickness of the dorsal wall and at least 50/50 heel to toe base of support (Figure 5) [22]. 

As all horses in this study presented with a grade 2 or above NPLA, remedial heel support was necessary. The choice of prosthetic was determined for each horse depending on the cost for the owner. Formahoof (www.formahoof.com) (*n* = 2; horses 1 and 2), wedge pads (www.3dhoofcare.com) (*n* = 3; horses 4, 5 and 6) or graduated duo ellipse (Hand made by YS) (*n* = 2; horses 3 and 7), hinds were used, with frog support padding, to provide elevation to establish an improved hoof pastern axis [22]. After shoeing with mechanically correct proportions and P3 elevation to within normal limits, the standing posture was re-photographed according to the previous method. 

The measurements taken from the photographs were an adapted Mawdsley score [13] for the point of buttock plumbline (POBPL) (Figure 6) and the pastern to dorsal wall angle difference axis (P-DW).

Metatarsal angle (MA) was taken from a line created from the bony ridge of the metatarsal, visible at the most proximal and distal ends of the metatarsal, hock angle was taken from the intersection of the dorsal surface of the metatarsal and the dorsal surface of the tarsus (HA), and coronet angle (CA) was measured taking a line through the most caudal and dorsal points of the hairline at the coronet band. The pictures were uploaded into Microsoft PowerPoint (Microsoft 365 MSO; Version 2210) and lines were put onto the images manually following parallel to the relevant anatomical points (Fig 3). A protractor was used to measure HA and P-DW. An angle measuring app able to measure angles with an accuracy of ±0.2° to ±0.3° was used to measure MA (Angle pro, iPhone 11). This was repeated on both sides of the horse.

Plantar angle (PA), proximal phalanx to middle phalanx angle difference (P1-P2), middle phalanx to distal phalanx angle difference (P2-P3), and P-DW were measured from lateral radiographs. The radiographs were uploaded into Microsoft PowerPoint, the centre of rotation (COR) of each joint was marked, the axis of each bone was drawn through the COR and the difference in the angle of the axis was collected [28] (Figure 7). The P-DW for the post shoeing group was measured using lateral photographs.

The numerical data obtained were subjected to the Anderson Darling normality test to determine if they were normal or non-parametric. Wilcoxon signed-rank tests were used to establish significant differences pre and post farriery intervention. A Pearson′s correlation test was used to measure the strength of the relationships between variables in the larger dataset. The alpha value was set at *p* = 0.05.

## 3. Results

In the pre intervention group, radiographically, the hinds presented with a larger difference in p2-p3 angle difference than p1-p2. The mean pre intervention p1-p2 angle difference for the right was −10.5° ± 3.1, the left was −9.9° ± 2.7. The mean pre intervention angle for p2-p3 angle difference was −13.9° ± 4.5 for the right and −16.5° ± 5.6 for the left (Table 3) 

Seven of the initial twelve horses has complete sets of measurements pre and post farriery intervention. Within this group, there was a significant difference in average POBPL pre and post intervention for both the right and left hinds (Table 4). The mean POBPL Mawdsley score when presenting with NPLA was 2.4 ± 0.5 in the right hind and 1.9 ± 0.3 in the left, after intervention these changed to 3.7 ± 0.9 and 3.6 ± 0.7, respectively. There was a significant positive correlation between POBPL and MA for both the right (*p* = 0.000) and left (*p* = 0.000). There was a significant difference in CA pre and post intervention for the right but not the left hind. The mean CA pre intervention was 29.7° ± 5.3 for the right and 31.3° ± 3.7 degrees for the left, post intervention averages were 22.7° ± 3.2 and 23.6° ± 3.2, respectively. There was a significant positive correlation between CA and P-DW on both the right (*p* = 0.001) and left (*p* = 0.004). There was a significant difference in MA pre and post intervention for both the right and left hind with the average for pre intervention being 81.3° ± 5.1 and post intervention being 88.0° ± 3.8 in the right hind and 74.4° ± 3.7 and 87.1° ± 2.9 in the left hind (Figure 7).

There was a significant difference in HA pre and post intervention in right but not the left hind. There was a significant difference in P-DW for both the right and left hind pre and post intervention (Figure 8).

There was a significant negative correlation between P-DW and POBPL on both the right (r = 0.622, *n* = 12, *p* = 0.004) and the left hinds (r = 0.733, *n* = 12, *p* = 0.000). There was a significant negative correlation between *p*-DW and MA for both the right (r = 0.470, *n* = 12, *p* = 0.043) and left (r = 0.740, *n* = 12, *p* = 0.000). There was a negative correlation between PA and MA for the right (r = −0.651, *n* = 12, *p* = 0.042) but a positive non-significant correlation for the left (r = 0.546, *n* = 12, *p* = 0.102). There was a significant negative correlation between PA and P-DW on the right (r = 0.826, *n* = 12, *p* = 0.003) but not on the left (r = −0.270, *n* = 12, *p* = 0.450). 

## 4. Discussion

This pilot study hypothesized that horses presenting with NPLA would also present with ACP and sought to isolate hoof balance as an influential factor in hind limb posture. It sought to begin to quantify ACP and the hoof imbalances associated. All the horses in this study (*n* = 12) presenting with NPLA presented with ACP, which was altered after farriery intervention (*n* = 7) (Figure 9). 

While previous studies discussed the biomechanical implications of NPLA possibly being responsible for the link to higher pathologies [2,5], this study suggests further research into posture as the link is warranted, not only as the suggested possible cause of NPLA [5], but also as the cause for increased and abnormal static and cumulative loads on the musculoskeletal system. This study suggests MA in association with POBPL as a measure of severity of ACP. While Mawdsley [13] was useful for grouping the animals, the addition of the MA created data with a normal distribution, enabling statistical analysis for posture as a variable in this pilot study and future research.

Anecdotally, sighting CA to the fore limb has been used as a gauge for PA, this study showed there was no significant correlation between the PA and CA, although the photos were focused on the hind end to address parallax so testing this theory was not possible. However, there was a strong correlation between P-DW and CA, showing CA as a measure of digit alignment. Considering the changes in limb position with hind hoof balance, shown by this study, the accuracy of sighting CA to the forelimb would depend on limb position, therefore actual CA could be more accurate in suggesting PA. This study found a difference in average CA pre and post intervention of 31.3° ± 3.7 and 23.6° ± 3.2 in the left hind with similar findings in the right hind, although the changes were not statistically significant in the right hind. Recent unpublished research [28] found that 90% of horses with a 30° CA and 88% of horses with a 28° CA presented with NPLA. In Practice, CA taken as a base line and sighted to the front carpus could be useful to document changes in hoof proportions along with P-DW (Figure 10). 

There was also a negative correlation between PA and MA for the right, but a positive non-significant correlation for the left, but there was a significant negative correlation between P-DW and MA, meaning that the data may suggest that P-DW is a more significant measurement then the absolute PA for postural changes. Considering that normal PA is variable, anywhere between 2° and 10°, it could be suggested that although a horse that has a normal PA of 2°, absent of the morphological changes associated with NPLA, may have a −2° PA, pre-intervention, with a difference of −4°, this would cause less of an increase in tendon strain and therefore have a reduced effect on posture than a horse who has a normal PA of 10°, with a 1° PA, pre-intervention, having a difference of −9° from its normal. This study suggests that the amount of angle change from the horse physiological norm has a more proprioceptive effect than whether the PA is negative. 

Recent studies [2,5] have focused on the link specifically between NPLA and higher pathologies. This study could suggest that future research into the link between hoof balance, posture, and pathologies in the hind limb and trunk, should perhaps include different severities of broken back hoof pastern axis (BBHPA) as well as NPLA. This is also supported by the fact that the horses in studies which presented with NPLA [2,5] showed the same posture as the horses which presented with long toe low heels in varying degrees [4]. Although NPLA is an extreme of long toe low heel conformation, suggesting long term poor balance, researching the links with less severe imbalances could help to further elucidate the timeline of causation. 

It was also noted that one of the horses still presented with a broken P-DW, while the other subjects all presented with 1 degree or less P-DW. This subject also still presented with a MA closer to the average of the pre-treatment group. This also supports the suggestion that a BBHPA should be added to further studies to see if the negative correlation between P-DW and MA continues with a positive PA but BBHPA. 

Farriery intervention to address hoof proportions and create a positive PA had a significant effect on P-DW and MA. Knowing that hoof proportions affect the distal interphalangeal joint over the proximal interphalangeal joint [25,26], decreasing hoof angle and reducing PA, this supports the findings that hoof proportions directly affect P-DW as the largest change in angle, pre intervention, was found in the distal interphalangeal joint on the radiographs. However, the proximal interphalangeal joint also presented with a broken axis, it was also noted that the P1 angles were steeper than one would expect from a straight hoof pastern axis, although this is anecdotal. Considering previous findings of a change in centre of pressure location during a shoeing cycle, 60% of the calculated change [25], further research could quantify the difference between P1 angles in aligned and broken back hoof pastern axes to determine a possible postural adaption in the proximal interphalangeal joint as the mechanism for these findings. 

P-DW was negatively correlated with MA, this suggests that hoof balance directly affects digit alignment and subsequently limb orientation. However, the mechanism can only be assumed, as this does not prove primary causation, as the question remains whether NPLA is a result of excess heel load from ACP. Measuring alterations in regional hoof loading, due to ACP, this resulting in changes to hoof growth and appearance, was beyond the scope of this study. Further research to quantify changes in load on the hoof with changes in limb orientation is warranted to clarify the two-way relationship between hoof balance and limb orientation.

The farriery intervention outlined in this study created an improved P-DW by providing elevation while maintaining frog support, and further longitudinal studies could research the efficacy of frog support as both a prevention and treatment method. Hoof morphology is a factor of changes in magnitude and direction of forces acting on the foot [27], further studies could outline the changes in pressure distribution on the hoofs solar surface from the treatment protocol adopted in this study. Some cases from this dataset were followed anecdotally and showed genuine hoof capsule improvement with elevation, with shoe fit to establish 50:50 proportions around the COR and frog support padding. Further studies could test different treatment protocols to establish the importance of each component of the intervention.

All the horses in this study were shod. Previous studies have reported better hoof proportions in the barefoot population in their study [2] and grade 2 and above NPA was associated with excessive crushing of the heels [23]. It was noted in this current study that the horses were all grade 2 NPLA and above but does not compare a shod and unshod population, however many of the subjects presented with heels lower than the frog. Going barefoot has been suggested for the treatment of NPLA [22], with engagement of the frog, as a preliminary treatment for NPLA to re-establish the frog and heels on a level plane. It is therefore assumed and supported by this study that open heel shoes could exacerbate the degree of NPLA due to prolapse of the caudal hoof and the ability of the heels to become lower than the frog height.

It could be suggested, in accordance with theoretical reviews [11,12], that proprioception from the mechanoreceptors in both the hoof and the flexor tendon, in the post intervention group, were responsible for changes in limb orientation. However, studies have found that the location of initial contact, the centre of pressure at mid stance and the point of breakover was not affected by regional anaesthesia of the distal limb [29], questioning the importance of proprioceptive input from the feet, therefore we should be cautious in assigning postural changes and causation to hoof balance in isolation. 

Although there was a significant change in right HA, the changes were smaller than expected and there was no significant difference in left HA pre and post intervention, despite a significant negative correlation between P-DW and MA and significant changes in MA and POBPL. This also suggests, due to the reciprocal apparatus, that the stifle angle was not significantly changed. Although this study did not measure higher joint changes, this points toward the change in limb orientation coming from higher structures, likely the pelvis, sacro-illiac or lumbo-sacral joint (Figure 11). 

This could explain why, anecdotally, many horses with NPLA are suggested to have sacroiliac region pain. Considering preliminary findings which suggested a link between thoracolumbar extension and over-riding spinous processes [19], research into the effects of ACP on the thoracolumbar region could be warranted and explain the anecdotal link between NPLA and spinal pathology. Further research to measure hip extension, pelvic inclination and thoracolumbar extension, and the effects on the spine, associated with ACP are indicated.

This study suggests, in agreement with previous studies [4], that the links between NPLA and pathology in the hind limb and into the trunk of the horse could be more extensive than the current research suggests. These pathologies could be concurrent, with posture being an important factor. However, more research is needed to establish causation. Future research could quantify concurrent pathologies, considering posture, and make links further along the myofascial lines and test the suggestions of correlating ACP with neuro-dental and upper cervical issues [21]. Whether the mechanisms creating ACP are purely proprioceptive or also antalgic responses to concurrent pathologies also remains to be established. 

### Limitations

Although this study begins to outline some relationships between hind hoof balance and hind limb posture, the small number of cases may have affected the significance of certain findings, and further research with larger numbers is required for clarification. As a pilot study undertaken in the field, there were variables outside of the researchers’ control. The radiographs were already collected so correct image acquisition could only be assumed. Some radiographs did not have the metatarsal visibly vertical to have as a reference for correct limb position and stance can influence radiographic measurements of hoof balance [30]. Post intervention radiographs were not taken, so a positive plantar angle was assumed by external references. Post intervention radiographs could help to quantify further the effect of farriery to the internal measurements in future studies.

Previous studies measuring joint angles placed the horses against a marked wall and used skin markers to accurately measure angles in the limb and ensure exact distance from the animal [16]. The researcher was unable to do this in the field, which could have led to inaccuracies. Although the parallax issue stated in previous studies [16] was addressed by focusing the photography on the hind end, parallax issues are suspected in the distal measurements of the digit to include CA and P-DW. Future studies should take measurements from photos focused on the individual areas of interest. While the farriery interventions created a similar outcome of a straight hoof pastern axis and proportions around the centre of rotation of the distal interphalangeal joint, and the trimming protocol was repeatable, the package used varied between subjects according to cost. The impact on variation between subjects was assumed to be insignificant, due to the same outcome, however, further research could scrutinize different methods to measure their impact. Future research should utilize more precise techniques for data acquisition to improve accuracy and repeatability. Radiograph acquisition should be standardized and repeated before and after treatment, farriery intervention should be standardized, and photograph acquisition should utilize traditional kinematic research methods and more accurate digital measurement systems should be used for all measurements. 

## 5. Conclusions

The results of this study suggest that hoof balance in the dorso-plantar plane and farriery intervention to change hind hoof proportions and digit alignment have a direct influence on hind limb posture. Further research into posture as a factor in the link between NPLA and higher pathologies is warranted, to outline the pathogenesis of ACP, as the key to elucidating pathogenesis. In this study, hoof balance was shown to play a role in limb orientation, and this could prove to be an important factor in both the genesis and amelioration of the relationship. Further studies quantifying the changes in the direction and magnitude of forces on both the hoof and musculoskeletal system from ACP are needed. 

## Figures and Tables

**Figure 1 animals-12-03275-f001:**
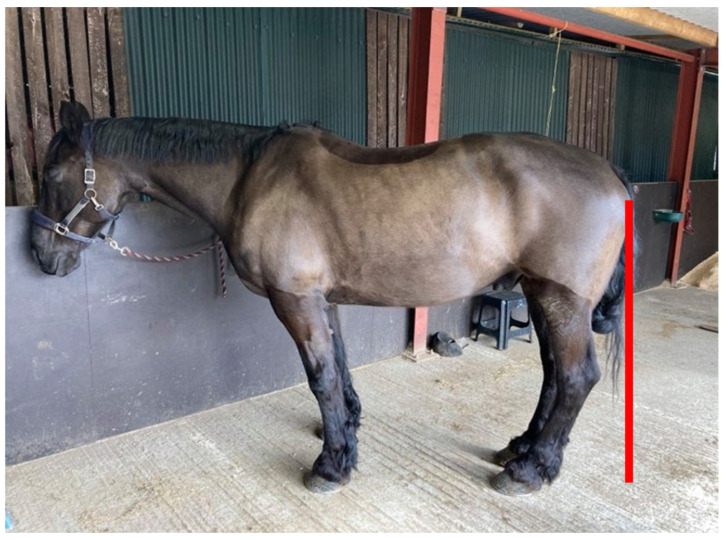
Canted-in posture, also defined as abnormal compensatory posture [10,11]. In this posture the limbs are brought closer together under the trunk of the horse creating a non-vertical metacarpal/tarsal alignment.

**Figure 3 animals-12-03275-f003:**
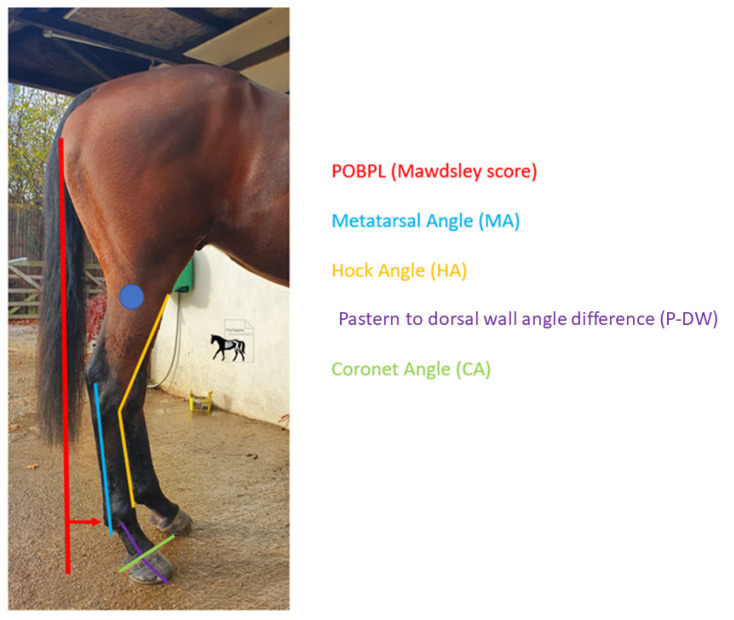
Digital photographs of the hind limb were taken at 3 metres, focused on the region of the distal biceps femoris/proximal extensor digitorum longus.

**Figure 4 animals-12-03275-f004:**
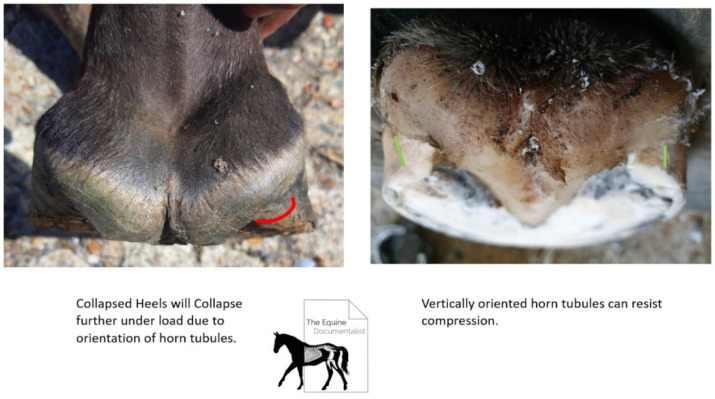
Heels were trimmed down until point of collapse, indicated by the point at which the tubules began to curl under. This created a vertical tubule orientation enabling the heel wall to cope with load.

**Figure 5 animals-12-03275-f005:**
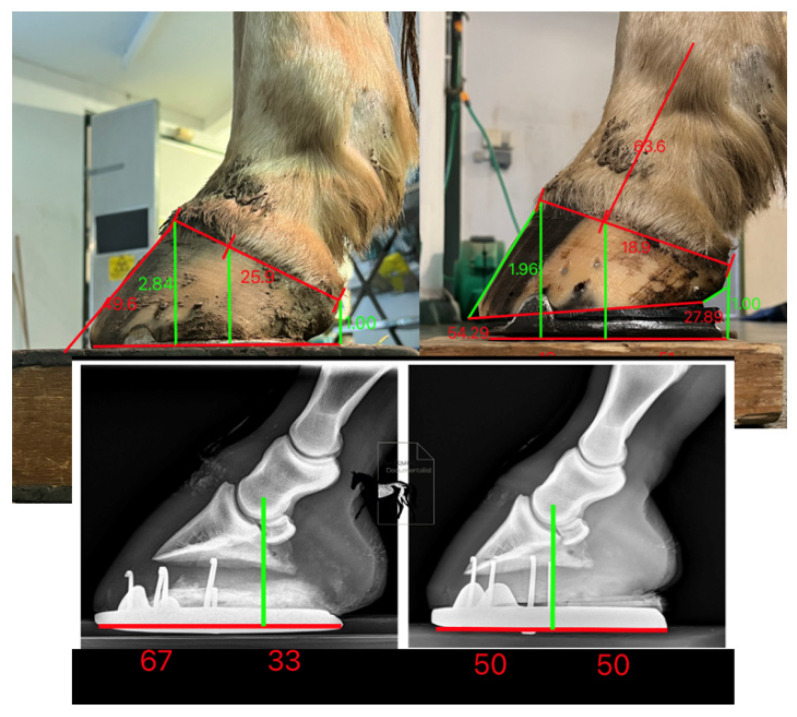
Shoeing protocol showing toe depth reduced and complimented with shoe fit and wedge pad elevation to create a positive plantar angle, with 50/50 proportions around the centre of rotation of the distal interphalangeal joint and an improved hoof-pastern axis.

**Figure 6 animals-12-03275-f006:**
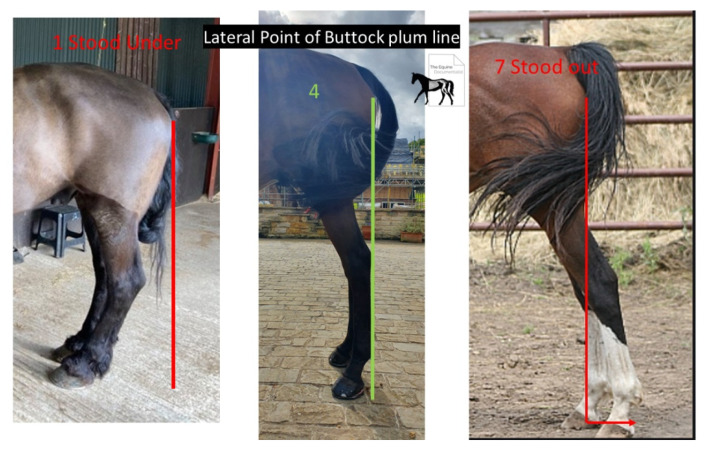
Mawdsley score for hind limb. The horses were given a Mawdsley et al. (1996) score for the point of buttock plumb line between 1 and 7, with a score of 4 being the ideal. This score and metatarsal angle were taken to ensure comprehensive understanding of postural changes. Authors’ own image.

**Figure 7 animals-12-03275-f007:**
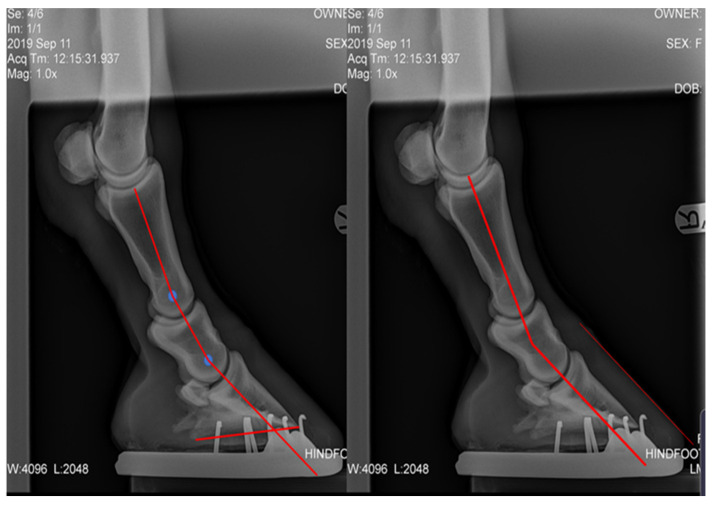
Two measurement analyses of the same pre-treatment radiograph showing the measurements taken from lateral radiographs. P1-P2, P2-P3, P-DW, and PA. Authors’ own image.

**Figure 8 animals-12-03275-f008:**
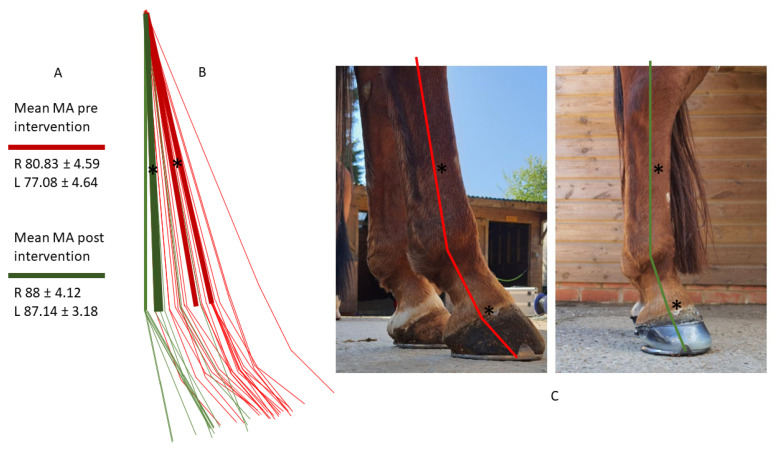
(**A**) Analysis showing mean MA pre and post intervention with standard deviation. (**B**) Graphic showing significant differences pre and post intervention of MA as denoted by *. Red lines show MA, pastern angle and hoof angle pre intervention, dark red lines show mean MA. Green lines show MA, pastern angle and hoof angle post intervention, dark green lines show mean post intervention MA. (**C**) Photo showing the lines in vivo with significant differences in MA and P-DW denoted by *.

**Figure 9 animals-12-03275-f009:**
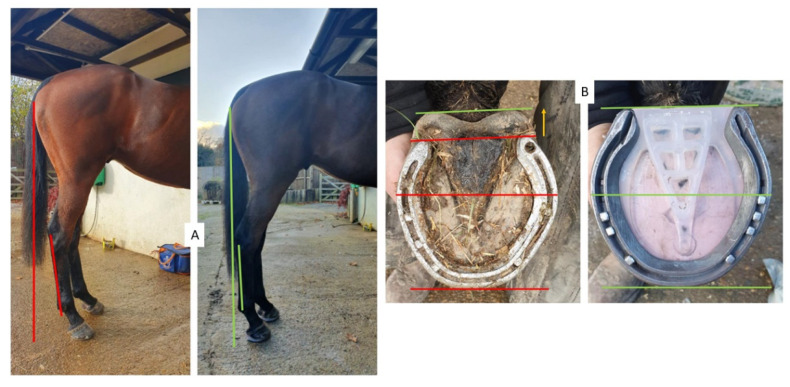
Change in POBPL and MA from changes in hoof proportions. (**A**) Improvements in POBPL and MA with the improvement of P-DW from farriery interventions (left-hand image—before; right hand image—after). (**B**) Improved toe to heel base of support ratios (left-hand image—before and right-hand image—after). Yellow arrow and green line show new heel base of support change. Red lines denote before, green lines denote after. Authors’ own images.

**Figure 10 animals-12-03275-f010:**
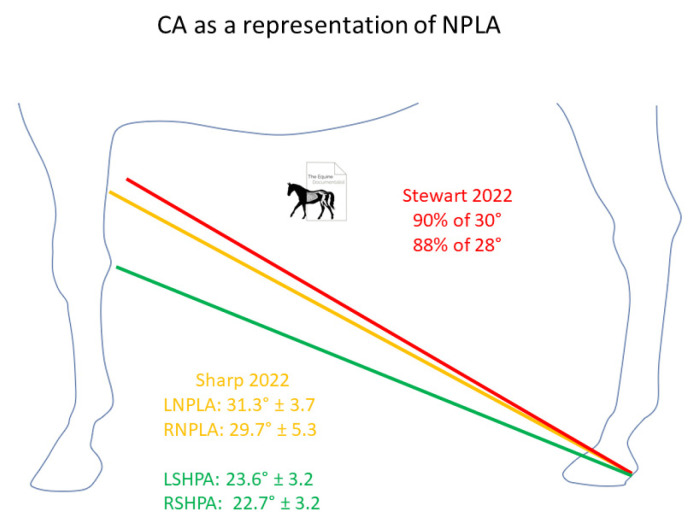
The findings of this study and previous unpublished research suggest that a CA of over 28° could suggest NPLA and recommendation for radiographs. Red shows findings of recent unpublished research. Yellow shows this current study′s pre-intervention averages and green shows post-intervention averages.

**Figure 11 animals-12-03275-f011:**
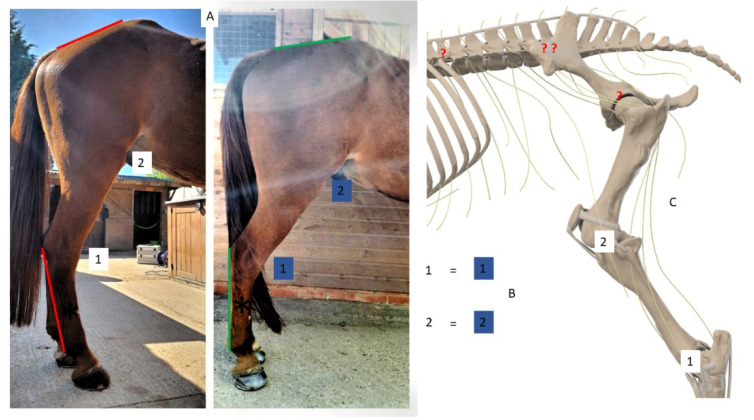
(A,B,C) Results of joint angle changes and suggested further study. A is a photo showing a pre and post intervention case. Red lines denote noted changes in angle with the significant finding of the change in MA denoted by *. Measuring changes in pelvic inclination (red and green lines along buttocks) was beyond the scope of this study but is suggested for further research. (B) According to the rule of the reciprocal apparatus, the non-significant change in HA informed the assumption of a non-significant change in stifle angle, shown as 1 = 1 and 2 = 2. C shows joints for further research, to elucidate where the change in limb posture is coming from, denoted by red question marks (sacro-illiac, lumbo-sacral, hip joint). The thoracolumbar joint is also highlighted as an area for research on further links.

**Table 1 animals-12-03275-t001:** Four-tier grading system for negative plantar angles (NPLA) [22], reproduced with permission, adapted from NPLA from negative palmar angles (NPA) [23].

Grade	Description
1—Mild	There is sufficient sole depth under the tip of the pedal bone, so trimming alone can restore a positive plantar angle.
2—Moderate	Sole depth at the toes is limited, so the best that can be achieved with trimming alone is a zero degree pedal bone angle. Further improvement to hoof-plantar angle requires mechanical shoeing
3—Severe	Heels, bars, digital cushion and bulbar cushion all compromised. Absence of complete heel mass under wings of the pedal bone. Positive plantar angle and/or digital alignment impossible without a special trim and artificial elevation.
4—Contracted	Severe heel collapse, with additional superficial digital flexor contracture giving them a very upright pastern angle. Vertical orientation of the proximal and middle phalanx and severe hyperextension of the distal interphalangeal joint.

**Table 2 animals-12-03275-t002:** Information on the horses used within this study.

Horse	Breed	Sex	Age (Years)
1	Thoroughbred—part bred	Gelding	5+
2	Thoroughbred	Gelding	5+
3	Irish Sport Horse	Gelding	5+
4	Thoroughbred	Gelding	13
5	Irish Sport Horse	Gelding	12
6	Thoroughbred—part bred	Mare	9
7	Irish Sport Horse	Mare	16
8	Thoroughbred—part bred	Mare	5+
9	Thoroughbred	Gelding	8
10	Thoroughbred	Gelding	12
11	Thoroughbred	Gelding	7
12	Thoroughbred—part bred	Gelding	20

**Table 3 animals-12-03275-t003:** Measurement data from pre-showing intervention radiographs of hind limbs (R = right hind; L = left hind; PA = plantar angle; P1 = proximal phalanx; P2 = middle phalanx; P3 = distal phalanx; Grade = negative plantar angle grade as per the 4 tier grading system for NPLA. Adapted for NPLA [2] from NPA [23]).

Horse	R PA	L PA	L P1-P2	R P1-P2	L P2–P3	R P2-P3	Grade
1	−1.5	−1.5	−12	−8	−18	−18	2
2	−5	−1	−8	−10	−22	−18	2
3	2	−4	−5	−5	−13	−17	2
4	1	−4	−9	−7	−23	−8	3
5	−8	−8	−11	−15	−23	−11	3
6	−3	−6	−12	−11	−17	−17	3
7	−4	−6	−14	−12	−18	−13	3
8	−2	−6	−12	−15	−7	−6	3
9	−2	−2	−10	−12	−7	−11	2
10	−5	−1.5	−6	−10	−17	−20	2
Mean	−2.75	−4	−9.9	−10.5	−16.5	−13.9	2.5
SD	−2.8	−2.3	−2.7	−3.1	−5.6	−4.5	0.5

**Table 4 animals-12-03275-t004:** Measurement data pre and post shoeing intervention from right and left hindlimbs.

Horse	POBPL	POBPL	MA	MA	HA	HA	P-DW	P-DW	CA	CA
	Right	Left	Right	Left	Right	Left	Right	Left	Right	Left
1_pre	2	2	73	71	143	150	20	19	34	28
2_pre	3	2	88	78	156	155	22	21	38	35
3_pre	2	2	77	77	151	151	12	12	27	35
4_pre	2	2	77	77	152	151	29	8	25	35
5_pre	3	1	85	67	152	155	19	26	35	32
6_pre	3	2	85	76	149	153	30	21	24	25
7_pre	2	2	84	75	151	150	22	21	25	29
Pre_Mean ± SD	2.4 ± 0.5 ^a^	1.9 ± 0.3 ^a^	81.3 ± 5.1 ^a^	74.4 ± 3.7 ^a^	150.6 ± 3.7 ^a^	152.1 ± 2.0	22.0 ± 5.7 ^a^	18.3 ± 5.7 ^a^	29.7 ± 5.3	31.3 ± 3.7 ^a^
1_post	5	5	90	90	151	151	0	0	21	21
2_post	4	4	90	90	159	156	0	0	22	24
3_post	2	3	79	86	155	155	0	0	21	25
4_post	3	3	87	83	155	152	18	10	30	30
5_post	4	3	90	83	151	151	0	0	23	23
6_post	4	4	90	90	158	158	1	1	19	19
7_pos	4	3	90	88	154	150	0	0	23	23
post_mean ± SD	3.7 ± 0.90 ^a^	3.6 ± 0.7 ^a^	88.0 ± 3.8 ^a^	87.1 ± 2.9 ^a^	154.7 ± 2.9 ^a^	153.3 ± 2.9	2.7 ± 6.2 ^a^	1.6 ± 3.5 ^a^	22.7 ± 3.2	23.6 ± 3.2 ^a^

(POBPL = point of buttock plumbline/Mawdsley score; MA = metatarsal angle in degrees; HA = hock angle in degrees; P-DW = pastern to hoof dorsal wall angle in degrees; CA = coronet angle in degrees). Superscript ‘a’ represents significant differences between pre- and post-intervention mean ± standard deviation (SD).

## Data Availability

Not applicable.

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
