# Peer review of "An Investigation into the Effects of Changing Dorso-Plantar Hoof Balance on Equine Hind Limb Posture"

_animals, 2022, doi:10.3390/ani12233275_

Round 1
Author Response
Thank you very much for your time reviewing our article, please see the attachment for our responses.
Reviewer 2 Report
Farrier science is a relevant aspect of the sport horse industry, and yet, research in this area is limited. The focus of dorso-plantar balance holds particular interest due to it's impact on soundness. Despite the importance of the topic, there are some limitations to address as it makes it difficult to review the manuscript in it's current format.
While the title refers to the work as a "pilot study", it appears almost to be a series of case studies performed by a farrier(s) treating balance issues within clients' horses. Was the farrier work done the same on all horses despite their conformation, health, and client preferences? Was the study blinded? Was the treatment replicated exactly for each subject? Even a hypothesis statement would be helpful to reflect whether this was just farrier work that was recorded or truly a research study.
More details are needed concerning methods. A clear indication of the number of horses utilized for the study needs to be present. Were some of the horses' data not utilized? Any inclusion/exclusion criteria? A table indicating information about these horses should be included that has age, weight, height, gender, and breed type. Clinical history of the horses with lameness evaluation should be given. Clearly indicate what ethical standards of evaluation was performed concerning the humane use of the animals for this research.
As for treatments, specifics on the shoeing methods needs to be given for each horse including the type of shoe and specifics about any adaptations made from prior studies and between horses used in this study. Was it the same for all? Author mentions differences in prosthetics/ wed pads, but how different and which horses were similar and which were not? For research purposes, costs restrictions due to client selection is not usually a consideration unless it is a case study. How much did cost consideration impact variation between subjects? Which horses were the heels removed including how much heel? What was the timing between measures including radiographs and photographs and between the farrier work and between treatment of each horse?
As for quantification methods, indicate information about the camera positioning for the photos including the distance, angle, height, and zoom. Who graded NPLA and what were their qualifications? Was it the same person using the same scoring system? Were reference markers used to indicate anatomical landmarks for photographs and if not how was consistency along with accuracy ensured? What was the angle app used? How accurate is that app and how does it work? Why not traditional kinematic measurement systems? Details concerning the computer programs used need to be included. Who did the radiographs and was it the same person? What were their qualifications? Were the radiographs taken in the same manner between horses, and how was that ensured?
As for the results, data needs to be reported using a table, and since the sample size is so small, data for each horse needs to be given and divided up between that taken using the app versus that taken from radiographs. The discussion seems to further explain the results, but the information is hard to follow as it seems divided between the two sections. Combining the results and discussion and adding in subsections where it clearly indicates what measurements that are being evaluated would be useful in making the results easier to follow. There appears to be a lot of statistical analysis for a pilot study, but without seeing the raw data, it's hard to distinguish what weight can be given concerning the findings, thus, tables with the exact measurements are needed. In addition, at times the discussion seems to be redundant readdressing information within the results, and thus, combining the two may assist in streamlining this aspect of the manuscript as the discussion seems lengthy at times. Finally, while the discussion needs to be reduced, the final aspect on limitations needs to be further explored.
Author Response

(The authors gave the same response as above.)

Round 2
Reviewer 2 Report
Authors are commended on the thorough revisions that they performed. While there are limitations with the methods of this study that could not be rectified without redoing the study, the revisions assist in clarifying these limitations and directing the reader to where value of the study needs to be taken. A few additional revisions are recommended. While an ethical statement is given at the end of the manuscript, at the beginning of the methods section the authors should add in the information that the authors had given the reviewer within their responses concerning review of the study:
The proposal for the study received ethical review by a university institution prior to the farriery work being undertaken. The horses were referred for intervention by their veterinary surgeons.
While the authors indicate in their responses of the review that no table is given concerning information specific to the horses due to the authors not having that information, the authors are still encouraged to include a table with this information on the horses within this study, even if the information is limited. Authors should reach out to the veterinarians and farriers that assisted with the study to get what information is available including age, breed, and gender. Also within the methods section indicate the PowerPoint version that was utilized. For figure 5, the picture on the left needs to be replaced with a clearer photo due to issues with glare.
In the results and discussion sections, table 3 is hard to follow. Adding lines separating out the pre- means/sd and the post- means/sd from the other numbers may be helpful. Instead of including the p-values within that table, since they were not exact numbers given, use superscripts to indicate significance related to p-value. For figure 8, the horse on the right needs to be without a blanket so that the conformation of the horse is better seen and a photo with less glare is also needed. Same for figure 10, as the glare for the middle photo makes it hard to clearly see the horse's conformation. Finally, in line 423, capitalize "future" at the beginning of the sentence.
Author Response
Thank you for your time reviewing the revisions to our paper. We have made further revisions based on your suggestions which are marked in red, within the manuscript and noted below.
Authors are commended on the thorough revisions that they performed. While there are limitations with the methods of this study that could not be rectified without redoing the study, the revisions assist in clarifying these limitations and directing the reader to where value of the study needs to be taken. A few additional revisions are recommended. While an ethical statement is given at the end of the manuscript, at the beginning of the methods section the authors should add in the information that the authors had given the reviewer within their responses concerning review of the study: The proposal for the study received ethical review by a university institution prior to the farriery work being undertaken.  The horses were referred for intervention by their veterinary surgeons.
Thank you for this suggestion and we agree it is important to highlight and therefore this statement has now been added to the main portion of the manuscript
While the authors indicate in their responses of the review that no table is given concerning information specific to the horses due to the authors not having that information, the authors are still encouraged to include a table with this information on the horses within this study, even if the information is limited. Authors should reach out to the veterinarians and farriers that assisted with the study to get what information is available including age, breed, and gender.
We have been able to gather most of the information you requested to be included and have now added it into a table within the methods section.
Horse |
Breed |
Sex |
Age (years) |
1 |
Thoroughbred - part bred |
Gelding |
5+ |
2 |
Thoroughbred |
Gelding |
5+ |
3 |
Irish Sport Horse |
Gelding |
5+ |
4 |
Thoroughbred |
Gelding |
13 |
5 |
Irish Sport Horse |
Gelding |
12 |
6 |
Thoroughbred - part bred |
Mare |
9 |
7 |
Irish Sport Horse |
Mare |
16 |
8 |
Thoroughbred - part bred |
Mare |
5+ |
9 |
Thoroughbred |
Gelding |
8 |
10 |
Thoroughbred |
Gelding |
12 |
11 |
Thoroughbred |
Gelding |
7 |
12 |
Thoroughbred - part bred |
Gelding |
20 |
Also within the methods section indicate the PowerPoint version that was utilized.
This has been added
For figure 5, the picture on the left needs to be replaced with a clearer photo due to issues with glare.
As suggested, we have altered this image for figure 5
In the results and discussion sections, table 3 is hard to follow. Adding lines separating out the pre- means/sd and the post- means/sd from the other numbers may be helpful. Instead of including the p-values within that table, since they were not exact numbers given, use superscripts to indicate significance related to p-value.
We have revised the tables with superscript letters and lines between cells to hopefully aid clarity of the information
For figure 8, the horse on the right needs to be without a blanket so that the conformation of the horse is better seen and a photo with less glare is also needed.
The has now been changed
Same for figure 10, as the glare for the middle photo makes it hard to clearly see the horse's Conformation.
We have tried to remove as much of the glare a possible but wish to keep this image as it demonstrates the effect on pelvis inclination as well and hind limb posture.
Finally, in line 423, capitalize "future" at the beginning of the sentence.
Now corrected thank you.